# Credit Attribution and Stable Compression

**Roi Livni**[*]          **Shay Moran**[†]          **Kobbi Nissim**[‡]          **Chirag Pabbaraju**[§]

## Abstract

Credit attribution is crucial across various fields. In academic research, proper citation acknowledges prior work and establishes original contributions. Similarly, in generative models, such as those trained on existing artworks or music, it is important to ensure that any generated content influenced by these works appropriately credits the original creators.

We study credit attribution by machine learning algorithms. We propose new definitions–relaxations of Differential Privacy–that weaken the stability guarantees for a designated subset of $k$ datapoints. These $k$ datapoints can be used non-stably with permission from their owners, potentially in exchange for compensation. Meanwhile, each of the remaining datapoints is guaranteed to have no significant influence on the algorithm's output.

Our framework extends well-studied notions of stability, including Differential Privacy ($k = 0$), differentially private learning with public data (where the $k$ public datapoints are fixed in advance), and stable sample compression (where the $k$ datapoints are selected adaptively by the algorithm). We examine the expressive power of these stability notions within the PAC learning framework, provide a comprehensive characterization of learnability for algorithms adhering to these principles, and propose directions and questions for future research.

## 1   Introduction

Many tasks that use machine learning algorithms require proper *credit attribution*. For example, consider a model trained on scientific papers that needs to reason about facts and figures based on existing literature. Most academic literature is protected under copyright licenses such as CC-BY 4.0 which allows adapting, remixing, transforming, and to copy and redistribute in any medium or format, as long as attribution is given to the creator. In another setting, a learner generating content, such as images or music, may benefit from creating *derivative works* from copyrighted materials without violating the creator's rights (either through proper attribution or monetary compensation, depending on the context and licensing).

The increasing use of ML algorithms and the need for greater transparency is reflected by the recently implemented EU AI Act, which mandates the disclosure of training data [14]. However, disclosure of training data and proper attribution are not necessarily equivalent. In particular, mere transparency of the dataset does not reveal whether certain elements of certain content have been derived, nor does it provide proper attribution when particular content is heavily built upon. Therefore, there is a need to develop more nuanced notions and definitions that enable learning under the constraint that works are properly attributed. This paper focuses on this challenge, exploring theoretical models of *credit attribution* to provide rigorous and meaningful definitions for the task.

---

[*]Tel Aviv University. `rlivni@tauex.tau.ac.il`.

[†]Technion and Google Research. `smoran@technion.ac.il`.

[‡]Georgetown University and Google Research. `kobbi.nissim@georgetown.edu`.

[§]Stanford University. `cpabbara@cs.stanford.edu`.

38th Conference on Neural Information Processing Systems (NeurIPS 2024).

Credit attribution is part of a much larger problem of learning under *copyright* constraints. Copyright issues in machine learning models are becoming increasingly prominent as these models are trained on vast amounts of data, often some of which is copyrighted. Consequently, the resulting models might contain content from copyrighted data in their training sets. Previous work suggests it may be mathematically challenging to capture algorithms that protect copyright. Specifically, attempts to regulate copyright often focus on protecting against *substantial similarity* between output content and training data by, for example, employing stable algorithms that are not sensitive to individual training points [9, 22, 24]. This is an important aspect of copyright; however, substantial similarity is only one piece of the puzzle.

Another piece of the puzzle involves the use of original elements from copyrighted works in a legally permissible manner, such as through *de minimis quotations*, transformative use, and other types of *fair uses*, such as learning and research [13]. To fully utilize ML in many practical scenarios, it is desirable for learning models to be allowed to use original elements in a similar manner.

To address this second piece, we focus on designing algorithms that, while allowed to use and be influenced by copyrighted material, must provide proper attribution. Such models would enable users to inspect these influences and verify that they conform to legal standards, or take necessary measures (such as monetary compensation or requesting permission). Despite credit attribution being narrower in scope than copyright protection in general, even this concept may be nuanced to be captured mathematically. Therefore, we focus on formalizing a specific (but arguably basic) aspect of it – *counterfactual attribution*:

> This principle asserts that any previous work that influenced the result should be credited. Counterfactually, if the creator of a work $W$ does not acknowledge another work $W'$, they should be able to produce $W$ as if they had no knowledge of $W'$.

For example, an argument based on this principle in the extreme case when $W = W'$ is found in a U.S. Supreme Court opinion:

> *". . . a work may be original even though it closely resembles other works, so long as the similarity is fortuitous, not the result of copying. To illustrate, assume that two poets, each ignorant of the other, compose identical poems. Neither work is novel, yet both are original. . . "*
>
> — *Feist Publications, Inc. v. Rural Telephone Service Company, Inc. 499 U.S. 340 (1991)*

## 2 Definitions and Examples

In this section, we introduce the two main definitions we study.

We first recall some standard notation from learning theory and differential privacy. Let $\mathcal{Z}$ be an input data domain and $\mathcal{C}$ denote an output space. We denote by $\mathcal{Z}^\star$ the set of all finite sequences with elements from $\mathcal{Z}$. Two sequences $S', S'' \in \mathcal{X}^\star$ are called neighbors if they have the same length $|S'| = |S''|$ and there is a unique index $i$ such that $S'_i \neq S''_i$. Let $\varepsilon, \delta > 0$ and let $p, q$ be probability distributions defined over the same space. We let $p \approx_{\varepsilon,\delta} q$ denote the following relation: $p(E) \leq \exp(\varepsilon) \cdot q(E) + \delta$ and $q(E) \leq \exp(\varepsilon) \cdot p(E) + \delta$ for every event $E$.

**Algorithms with Credit Attribution.** Consider a mechanism $M : \mathcal{Z}^\star \to \mathcal{C} \times \mathcal{Z}^\star$ that, for every possible input sequence $S = (z_1, \ldots, z_n)$, outputs a pair $(c, R)$, where $c \in \mathcal{C}$ and $R \in \mathcal{Z}^\star$. Intuitively, $R$ is the list of inputs being credited by the mechanism, and $c$ is the model/content produced by the mechanism. Thus, we require that each data point $z_i \in R$ is also an input data point $z_i \in S$. For such a mechanism $M$ and an input sequence $S$, we let $M(S)$ denote the probability distribution over outputs of $M$ given $S$ as input, where the probability is induced by the internal randomness of the mechanism. For example, if $M$ is deterministic, then $M(S)$ is a Dirac distribution (i.e., it assigns probability 1 to the deterministic output of $M$ on $S$).

The definition below uses the following notation: for a sequence $S = (z_1, \ldots, z_n)$ and an index $i \in [n]$, we let $S_{-i}$ denote the subsequence of $S$ obtained by omitting $z_i$. Let $z_i \in S$ be a data point such that $\Pr_{(c,R) \sim M(S)}[z_i \in R] < 1$. That is, there is a positive probability that $z_i$ is not credited

**Example: Support Vector Machine**

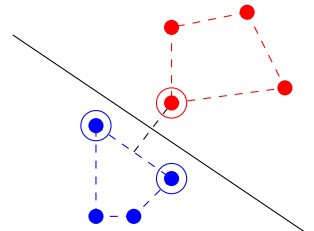

Figure 1: Support Vector Machine (SVM) as an ($\varepsilon = \delta = 0$)-counterfactual credit attributor: The SVM algorithm identifies a maximum-margin separating hyperplane, which is defined by the subsample of the support vectors. Any input point which is not a support vector does not influence the output: even if it is removed from the input sample, the output hyperplane does not change.

by $M$ when executed on $S$. In this case, we let $M(S^{-i})$ denote the distribution of $M(S)$ conditioned on the event that $z_i \notin R$. We are now ready to present our first definition[5] of counterfactual credit attribution.

---

**Definition 1** (Counterfactual Credit Attribution). *Let $\varepsilon, \delta > 0$. A mechanism $M : \mathcal{Z}^\star \to \mathcal{C} \times \mathcal{Z}^\star$ is called an ($\varepsilon, \delta$)-counterfactual credit attributor (CCA) if for every input sequence $S = (z_1, \ldots, z_n)$ and every index $i \in [n]$ the following holds: either $\Pr_{(c,R) \sim M(S)}[z_i \in R] = 1$, or*

$$M(S^{-i}) \approx_{\varepsilon, \delta} M(S_{-i}),$$

*where $M(S_{-i})$ is the output distribution on the dataset $S_{-i} = S \setminus \{z_i\}$, and $M(S^{-i})$ is the output distribution on the dataset $S$, conditioned on $z_i \notin R$.*

---

To emphasize, in Definition 1 the conditional output distribution $M(S^{-i})$ models the condition "*if data-point $z_i$ is not credited by $M$,*" whereas the output distribution $M(S_{-i})$ represents the counterfactual scenario "*had the data-point $z_i$ not been seen by $M$.*"

**Example 2.1** (Stable Sample Compression [8, 20]). A mechanism $A : \mathcal{Z}^\star \to \mathcal{C}$ is a stable sample compression scheme of size $k$ if for every input sequence $S = (z_1, \ldots, z_n)$ there is a subsequence $\kappa(S) \subseteq S$ of size $|\kappa(S)| \le k$ such that $A(S) = A(T)$ for every intermediate subsequence $\kappa(S) \subseteq T \subseteq S$. See Figure 1 for an example.

Each stable compression scheme corresponds to an ($\varepsilon = 0, \delta = 0$)-CCA which credits the datapoints in $\kappa(S)$. That is, $M(S) = (A(S), \kappa(S))$. Stable sample compression thus provides something stronger: group-counterfactuality, meaning any subset of datapoints that is not selected does not influence the output.

Definition 1 not only relaxes stable sample compression, but also extends the concept of differential privacy with public data, known as semi-private learning. In semi-private learning, the learner's input includes public examples (which can be processed non-stably) and private examples (for which the algorithm must satisfy differential privacy guarantees). Semi-private learning [1, 4] has been extensively studied in recent years [19], for example, in the context of query release [3, 18], distribution learning [5, 6], computational efficiency [7, 21], as well as in other contexts.

**Definition 2** (Semi-Differentially Private Mechanism). *Let $\varepsilon, \delta > 0$; an ($\varepsilon, \delta$)-semi differentially private (semi-DP) mechanism is a mapping $M : \mathcal{Z}^\star \times \mathcal{Z}^\star \to \mathcal{C}$ such that for every $S_{\mathrm{pub}} \in \mathcal{Z}^\star$ and every pair of neighboring sequences ,$S'_{\mathrm{priv}}, S''_{\mathrm{priv}}$:[6]*

$$M(S_{\mathrm{pub}}, S'_{\mathrm{priv}}) \approx_{\varepsilon, \delta} M(S_{\mathrm{pub}}, S''_{\mathrm{priv}}).$$

---

[5]In analogy to the variant of differential privacy where the unit of protection is the addition or removal of a data point, our definition uses omissions of data points. This aligns with our motivation for counterfactual credit attribution: if a non-credited data point is omitted (rather than replaced), the output does not change. Omission is crucial here, as replacing a non-credited data point with a credited one could drastically alter the output.

[6]Note that the special case of $S_{\mathrm{pub}} = \emptyset$ gives a DP mechanism.

**Remark 1.** Any semi-DP mechanism $M$ that uses $k$ public points can be turned into a CCA mechanism as follows: on an input sequence $S$, the CCA mechanism outputs $(c, R)$, where $R = S_{\leq k}$, and $c = M(S_{\leq k}, S_{>k})$. That is, $M$ uses the first $k$ points in $S$ as public data, and the rest are private.

Private learning with public data is sometimes likened to semi-supervised learning, where private data corresponds to unlabeled data and public data to labeled data. In both scenarios, the learner accesses many less informative examples (unlabeled or private) and fewer more informative examples (labeled or public). Expanding on this analogy, Definition 1 is akin to active learning, where the learner adaptively chooses which data points to credit, similar to selecting which data points to label in active learning.

Semi-differential privacy (Definition 2) provides stronger stability guarantees than counterfactual credit attribution (Definition 1), including for the selection process. In contrast, Definition 1 allows for a highly non-stable selection process (e.g., SVM). This leads us to consider a more direct hybrid of semi-DP and sample compression, suggesting the following definition:

---

**Definition 3** (Sample DP-Compression Scheme). *Let $\varepsilon, \delta \geq 0$ and $k \leq n$. An $(\varepsilon, \delta)$ sample differentially private $(n \to k)$-compression scheme is a mechanism $M : \mathbb{Z}^n \to \mathcal{C}$ which consists of two functions:*

1. ***Compression:*** *an $(\varepsilon, \delta)$-DP mechanism $\kappa : \mathbb{Z}^n \to [n]^k$, called the compression function, and*

2. ***Reconstruction:*** *an $(\varepsilon, \delta)$ semi-DP mechanism $\rho : \mathbb{Z}^\star \times \mathbb{Z}^\star \to \mathcal{C}$ called the reconstruction function.*

*Then, for every input sequence $S$:*

$$M(S) = \rho(S|_{\kappa(S)}, S|_{\neg\kappa(S)}),$$

*where $S|_{\kappa(S)} = (S_i)_{i \in \kappa(S)}$ and $S|_{\neg\kappa(S)} = (S_i)_{i \notin \kappa(S)}$.*

---

Note that the compression function $\kappa$ selects the indices of the compressed subsample (rather than the subsample itself, as in classical sample compression). This technical difference allows us to pose the requirement of differential privacy on the compression function $\kappa$. Going back to the analogy with active learning, Definition 2 also imposes stability of the labeling function (i.e. the function that decides which labels to query).

**Example 2.2** (Randomized Response). We next describe a simple task which can be performed by sample DP-compression schemes, but not by semi-DP mechanisms. Imagine that the data is drawn from a distribution where each datapoint is useful with probability 0.1 and is otherwise garbage with probability 0.9. The goal is to select $k$ datapoints while maximizing the number of useful datapoints that are selected. If we select datapoints obliviously, for example by simply taking the first $k$ examples, we would expect that only about 10% of them will be useful. However, by using a mechanism compliant with Definition 3, we can increase the proportion of useful examples.

This mechanism is based on randomized response and operates as follows: each example is independently assigned a random label in $\{0, 1\}$, where a useful example is assigned a label of 1 with probability $p > 1/2$, and each garbage example is assigned a label of 1 with probability $1 - p < 1/2$. The value of $p$ is set as a function of the privacy parameter $\varepsilon$.[7] Then, the compression function $\kappa$ selects the first $k$ indices whose label is 1. This way, the fraction of useful points among the points labeled 1 is $\approx \frac{0.1p}{0.1p + 0.9(1-p)} = \frac{1}{9/p - 8} > 0.1$ (the last inequality holds for $p > 1/2$). See Appendix B for a more detailed argument.

## 3  Main Theorems

In this section, we present our main theorems that characterize the expressivity of learning rules satisfying our proposed definitions. We focus on the PAC (Probably Approximately Correct) learning model [23] and employ its standard definitions (explicitly provided in Section 4).

---

[7]We get $\varepsilon = \ln\left(\frac{p}{1-p}\right)$ and $\delta = 0$.

**Question** (Guiding Question). Is learnability subject to counterfactual credit attribution (Definition 1) more restricted than unconstrained learnability? Is learnability subject to sample DP-compression (Definition 3) more restricted than unconstrained learnability? How do these restrictions compare to differentially private learning?

Note that with respect to both Definition 1 and Definition 3, it is clear that if $k$, the number of credited points, is sufficiently large, then it is possible to learn any PAC-learnable class $\mathcal{C}$. Indeed, if $k$ equals the PAC sample complexity of $\mathcal{C}$, then an oblivious selection, such as the first $k$ points, will suffice. Therefore, the above question is particularly interesting for values of $k$ that are significantly smaller than the PAC sample complexity of $\mathcal{C}$.

Our first theorem demonstrates that every PAC-learnable class can be learned using an $(\varepsilon = 0, \delta = 0)$-counterfactual credit attribution learning rule, which selects at most a logarithmic number of sample points for attribution. Remarkably, this can be achieved using the AdaBoost algorithm.

---

**Theorem 1** (PAC Learning with Credit Attribution = PAC Learning). *Let $\mathcal{C}$ be a concept class with VC dimension $\mathrm{VC}(\mathcal{C}) = d < \infty$, and let $\alpha, \beta$ denote the error and confidence parameters. Then, there exists an $(\varepsilon = 0, \delta = 0)$-CCA learning rule $M$ that learns $\mathcal{C}$ with sample complexity $n = O\left(\frac{d \log(d/\alpha) + d \log(1/\beta)}{\alpha}\right)$, while selecting only $k = O(d \log n)$ examples for attribution.*

---

We leave as an open question whether $k$ can be made independent of $n$, possibly by allowing $\varepsilon$ and $\delta$ to be positive. Note that an affirmative answer to this question might also shed light on the sample compression conjecture [17, 25].

Our second theorem establishes a limitation for sample DP-compression schemes, showing that they do not offer more expressivity than differentially private PAC learning [15].

---

**Theorem 2** (Sublinear Sample DP-Compression = DP Learning). *Every concept class $\mathcal{C}$ satisfies exactly one of the following:*

   *1. $\mathcal{C}$ is learnable by a DP-learner.*

   *2. Any sample DP-compression scheme that learns $\mathcal{C}$ has size at least $k = \Omega(1/\alpha)$.*

---

Theorem 2 implies a stark dichotomy: either a class $\mathcal{C}$ can be learned by a DP algorithm (equivalently, a sample DP-compression of size $k = 0$), or it is impossible to learn it unless $k = \Omega(1/\alpha)$. Notice that with $k = O(d/\alpha)$, public examples are sufficient to learn without any private examples. Theorem 2 generalizes a result by [1, Theorem 4.2] who proved it in the special case of semi-DP learning. In our setting though, we need to crucially handle scenarios where the credited (or rather, public) datapoints are chosen *adaptively* as a function of the full dataset. This is not the case in semi-DP learning, and requires us to use novel technical tools (like Theorem 3 ahead).

Thus, in the PAC setting, sample DP-compression schemes do not offer any advantage over semi-DP learners. However, Example 2.2 demonstrates that using sample DP-compression, it is possible to select the $k$ points in the compression set so that the frequency of 'useful' examples among these $k$ points is boosted.

Our next theorem addresses the limits of handpicking $k$ points by sample DP-compression. We formalize this task as follows: given a distribution $D$ over $\mathcal{Z}$ and an event $E$ of 'good' points, the goal is to design a DP-compression function $\kappa : \mathcal{Z}^n \to [n]^k$ that maximizes the number of selected data points that belong to $E$. That is, the goal is to maximize

$$\sum_{i \in \kappa(S)} 1[z_i \in E].$$

Example 2.2 illustrates a method that selects roughly $\exp(\varepsilon) \cdot k \cdot D(E)$ points from $E$ by an $(\varepsilon \geq 0, \delta = 0)$-compression function. This is a factor of $\exp(\varepsilon)$ better than obliviously selecting the $k$ points, which yields $k \cdot D(E)$ points from $E$. Is this factor of $\exp(\varepsilon)$ optimal? Can one do better, possibly by increasing $\delta$? The following result shows that $\exp(\varepsilon)$ is asymptotically optimal.

**Theorem 3.** *Let $M$ be an $(\varepsilon, \delta)$ sample DP-compression scheme, let $\mathcal{D}$ be a distribution over $\mathcal{Z}$, and let $E \subseteq \mathcal{Z}$ be any event, with $p = \mathcal{D}(E)$. For an input sample $S = (z_1, \ldots, z_n) \sim \mathcal{D}^n$, define $Z = Z(S)$ as the random variable denoting the fraction of selected indices in $\kappa(S)$ whose corresponding data points belong to $E$. That is, $Z = \frac{1}{|\kappa(S)|} \sum_{i \in \kappa(S)} 1[z_i \in E]$. Then,*

$$pe^{-\varepsilon} - \delta n \leq \mathbb{E}[Z] \leq pe^{\varepsilon} + \delta n. \tag{1}$$

Indeed, since by convention $\delta = \delta(n) \ll 1/n$, the above theorem implies that $\exp(\varepsilon)$ is asymptotically optimal. We note that Theorem 3 is also key in the proof of Theorem 2. We elaborate on this in Section 4.2.

**Generalization.** Definition 1 and Definition 3 can also be examined from a learning theoretic perspective as notions of algorithmic stability. Algorithmic stability is particularly useful in the context of generalization because, roughly speaking, stable algorithms typically generalize well. We note in passing that this is indeed the case for Definition 1 and Definition 3: any learning rule adhering to either definition satisfies that its empirical error and population error are typically close. One natural way to prove this is by following the argument that shows sample compression schemes generalize. In a nutshell, the argument proceeds as follows: first, if we fix the selected $k$-tuple, the obtained hypothesis generalizes well. Then, we apply a union bound over all possible $n^k$ choices of $k$-tuples from the input sample.

## 4 Technical Background and Proofs

We study our main definitions in the context of PAC learning. Concretely, we assume that the input data domain $\mathcal{Z}$ in Section 2 is $\mathcal{X} \times \mathcal{Y}$, for an input space $\mathcal{X}$ and label space $\mathcal{Y}$. For our purposes, $\mathcal{Y} = \{0, 1\}$. Learning rules are mechanisms $\mathcal{A} : \mathcal{Z}^* \to \mathcal{C} \times \mathcal{Z}^*$, where $\mathcal{C}$ is the set of all functions mapping $\mathcal{X}$ to $\mathcal{Y}$, denoted as $\mathcal{Y}^{\mathcal{X}}$. We say that a distribution $\mathcal{D}$ over $\mathcal{Z}$ is *realizable* by a hypothesis class $\mathcal{H} \subseteq \mathcal{Y}^{\mathcal{X}}$ if for every finite sequence $(x_1, y_1), \ldots, (x_n, y_n)$ drawn i.i.d from $\mathcal{D}$, there exists some hypothesis $h \in \mathcal{H}$ that satisfies $h(x_i) = y_i, \forall i \in [n]$. For any hypothesis $h \in \mathcal{Y}^{\mathcal{X}}$, we denote its risk with respect to a distribution $\mathcal{D}$ by $R_{\mathcal{D}}(h) = \Pr_{(x,y) \sim \mathcal{D}}[h(x) \neq y]$.

**Definition 4** (CCA PAC learning rule). *A mechanism $\mathcal{A}$ is a CCA PAC learning rule for a hypothesis class $\mathcal{H}$, if $\mathcal{A}$ satisfies Definition 1, and for any distribution $\mathcal{D}$ realizable by $\mathcal{H}$, for any $\alpha, \beta > 0$, there exists a finite $n = n_{\mathcal{A}}(\alpha, \beta)$, such that with probability at least $1 - \beta$ over a sample $S \sim \mathcal{D}^n$ and the randomness of $\mathcal{A}$, the hypothesis $h$ in the output $(h, S')$ of $\mathcal{A}$ on $S$ satisfies $R_{\mathcal{D}}(h) \leq \alpha$.*

**Definition 5** (Sample DP-Compression learning rule). *An $(\varepsilon, \delta)$ sample differentially private $(n \to k)$ compression scheme $M$ learns a hypothesis class $\mathcal{H} \subseteq \mathcal{Y}^{\mathcal{X}}$, if for any distribution $\mathcal{D}$ realizable by $\mathcal{H}$, for any $\alpha, \beta > 0$, with probability at least $1 - \beta$ over a sample $S \sim \mathcal{D}^n$ and the randomness of $M$, the hypothesis $M(S)$ output by the reconstruction function in $M$ satisfies $R_{\mathcal{D}}(M(S)) \leq \alpha$.*

**Remark 2.** Note that if $k = 0$ above, we recover the standard definition of an $(\alpha, \beta, \varepsilon, \delta)$-DP PAC learner (where $\alpha$ is the error, $\beta$ is the failure probability, and $\varepsilon, \delta$ are the privacy parameters) [15].

### 4.1 Upper Bound: PAC learnability implies $(\varepsilon = \delta = 0)$-counterfactual credit attribution learning

Our CCA learning rule crucially uses the notion of a *randomized* stable sample compression scheme, which is a generalization of stable sample compression schemes (Example 2.1) and was developed in a recent work by [12]. We use the notation $S' \sqsubseteq S$ for sequences $S, S' \in (\mathcal{X} \times \mathcal{Y})^*$ that satisfy: $(\forall(x, y)) : (x, y) \in S' \implies (x, y) \in S$.

**Definition 6** (Stable Randomized Sample Compression Scheme). *A randomized sample compression scheme $(\mathcal{D}_\kappa, \rho)$ for a class $\mathcal{H}$ having failure probability $\xi$ comprises of a distribution $\mathcal{D}_\kappa$ over (deterministic) compression functions $\kappa : (\mathcal{X} \times \mathcal{Y})^* \to (\mathcal{X} \times \mathcal{Y})^*$ and a deterministic reconstruction function [8] $\rho : (\mathcal{X} \times \mathcal{Y})^* \to \mathcal{Y}^{\mathcal{X}}$. The compression functions $\kappa$ in the support of $\mathcal{D}_\kappa$ must satisfy*

---

[8]It seems interesting to possibly consider randomized reconstruction functions as well; for our purposes, deterministic reconstruction functions suffice.

- *For any $S \in (\mathfrak{X} \times \mathcal{Y})^*$ realizable by $\mathcal{H}$, if $\kappa(S) = S'$, then $S' \sqsubseteq S$.*

*The reconstruction function $\rho$ must satisfy*

- *For any $S \in (\mathfrak{X} \times \mathcal{Y})^*$ realizable by $\mathcal{H}$,*

$$\Pr_{\kappa \sim \mathcal{D}_\kappa} [\exists (x, y) \in S : \rho(\kappa(S))(x) \neq y] \leq \xi. \tag{2}$$

*A randomized sample compression scheme $(\mathcal{D}_\kappa, \rho)$ for $\mathcal{H}$ is stable if for any $S \in (\mathfrak{X} \times \mathcal{Y})^*$ realizable by $\mathcal{H}$ and $S' \sqsubseteq S$, the distribution of $\kappa(S')$ is the same as the distribution of $\kappa(S)$ conditioned on $\kappa(S) \sqsubseteq S'$. The size $s(n)$ of the compression scheme is the supremum over $S \in (\mathfrak{X} \times \mathcal{Y})^n$ (realizable by $\mathcal{H}$) and $\kappa$ in the supportt of $\mathcal{D}_\kappa$ of the number of distinct elements in $\kappa(S)$.*

[12] show that stable randomized compression schemes imply generalization.

**Lemma 4.1** (Theorem 1.2 in [12])**.** *Let $(\mathcal{D}_\kappa, \rho)$ be a stable randomized compression scheme for $\mathcal{H}$ of size $s(n)$ and failure probability $\xi$. Let $\mathcal{D}$ be any distribution over $\mathfrak{X} \times \mathcal{Y}$ realizable by $\mathcal{H}$. For any $n$ and $\beta > 2\xi$, with probability at least $1 - \beta$ over $S \sim \mathcal{D}^n$ and $\kappa \sim \mathcal{D}_\kappa$, it holds that*

$$R_\mathcal{D}(\rho(\kappa(S))) \leq O\left(\frac{s(n) + \log(1/\beta)}{n}\right).$$

Furthermore, they also show that there exists a stable randomized compression scheme for any hypothesis class $\mathcal{H}$ having finite VC dimension $d$. This compression scheme is based on a simple variant of AdaBoost (Algorithm 1 in [12]). The following is contained in their work:[9]

**Lemma 4.2** ([12])**.** *For any hypothesis class $\mathcal{H}$ with VC dimension $d$, there exists a stable randomized sample compression scheme (based on AdaBoost) having failure probability $\xi$ of size*

$$s(n) = O\left(d \log(n/\xi)\right). \tag{3}$$

We are now equipped with the necessary tools required to prove Theorem 1.

*Proof of Theorem 1.* Let $\mathcal{D}$ be any distribution realizable by $\mathcal{H}$, and let $S$ be a sample of size $n$ drawn from $\mathcal{D}^n$. Given $\beta$, fix $\xi = \beta/3$. From Lemma 4.2, we know that there exists a stable randomized compression scheme $(\mathcal{D}_\kappa, \rho)$ for $\mathcal{H}$ of size $s(n) = O(d \log(n/\beta))$, and failure probability $\xi$. Then, since $\beta > 2\xi$, from Lemma 4.1, we know that with probability at least $1 - \beta$ over $S$ and $\kappa \sim \mathcal{D}_\kappa$,

$$R_\mathcal{D}(\rho(\kappa(S))) \leq O\left(\frac{d \log(n/\beta)}{n}\right).$$

For the right-hand size above to be at most $\alpha$, it suffices to have $n = O\left(\frac{d \log(d/\alpha) + d \log(1/\beta)}{\alpha}\right)$.

Let $\mathcal{A}$ be the learning rule, which when given a sample $S \sim \mathcal{D}^n$ as input, runs the stable randomized compression scheme from above on $S$ to obtain $S'$ of size $k = O(d \log(n/\beta))$. The learner then outputs $(\rho(S'), S')$. By the reasoning above, $\rho(S')$ has error at most $\alpha$ with probability at least $1 - \beta$.

It remains to argue that $\mathcal{A}$ is a valid CCA mechanism. This follows by virtue of $(\mathcal{D}_\kappa, \rho)$ being a *stable* randomized compression scheme. Namely, for any $i$, $S_{-i} \sqsubseteq S$, and hence by Definition 6, the distribution of $\kappa(S_{-i})$ is identical to the distribution of $\kappa(S)$ conditioned on $S_i \notin \kappa(S)$. Finally, since $\rho$ is a deterministic function of its argument, $\mathcal{A}$ satisfies Definition 1 with $\varepsilon = \delta = 0$. ∎

## 4.2 Lower Bound: A dichotomy for sample DP-compression

Towards proving Theorem 2, we first show that a sample DP-compression scheme for the class of *thresholds* can be used to construct a DP learner for it. This lemma has a similar flavor to the public data reduction lemma (Lemma 4.4) in [1]. For a set $S = \{x_1, \ldots, x_m\}$, the class of thresholds over $S$ comprises of $m$ functions $h_1, \ldots, h_m$ such that $h_i(x_j) = \mathbb{1}[i \leq j]$, $\forall i, j \in [m]$.

---

[9]In more detail, this follows by setting the weak learning parameter $\gamma$ to a constant (e.g., $1/8$) in Algorithm 1 in [12], and noting that such a weak learner can be found via empirical risk minimization.

**Lemma 4.3** (Reduction from DP learner to sample DP-compression scheme). *Let $\mathcal{H}_m$ be the class of thresholds over $\{x_1, \ldots, x_m\}$. Suppose there exists an $(\varepsilon, \delta)$ sample DP-compression scheme $\tilde{A}$ that learns $\mathcal{H}_m$ with error $\alpha$ and failure probability $\beta = \frac{1}{32}$, and has sample complexity n and compression size $k \leq n$. Let $\delta \leq \frac{1}{64n^2}$. Then, there exists a $\left(64ke^\varepsilon\alpha, \frac{1}{16}, 2\varepsilon, 3\delta\right)$-DP learner $\mathcal{A}$ for $\mathcal{H}_{m-1}$, where $\mathcal{H}_{m-1}$ is the class of thresholds over $\{x_1, \ldots, x_{m-1}\}$, with sample complexity n.*

*Proof.* Let $\mathcal{D}$ be any distribution over $\{x_1, \ldots, x_{m-1}\} \times \{0, 1\}$ realizable by $\mathcal{H}_{m-1}$. Given a sample $S \sim \mathcal{D}^n$, the private learner $\mathcal{A}$ does the following. First, it constructs a sample $\tilde{S}$, also of size $n$, as follows. Initialize $j = 1$. For each $i = 1, 2, \ldots, n$, toss a coin (independently of the data, and other coins) that lands heads with probability $p$, for $p$ to be appropriately chosen later. If the coin lands heads, $\tilde{S}(i) = S(j)$, and $j$ is incremented by 1. If the coin lands tails, $\tilde{S}(i)$ is set to the designated dummy example $(x_m, 1)$. In this way, $\tilde{S}$ is a sample of size $n$ drawn from the *mixture* distribution $\tilde{\mathcal{D}} = p \cdot \mathcal{D} + (1 - p) \cdot \mathbb{1}_{(x_m, 1)}$, where $\mathbb{1}_{(x_m, 1)}$ is a point mass on $(x_m, 1)$. Note that since all the thresholds in $\mathcal{H}_m$ label $x_m$ as 1, $\tilde{\mathcal{D}}$ is realizable by $\mathcal{H}_m$.

The learner $\mathcal{A}$ now invokes the sample DP-compression scheme $\tilde{A}$ on $\tilde{S}$. If *any* of the $k$ examples in the compression set constructed by $\tilde{A}$ is a non-dummy element, $\mathcal{A}$ outputs a constant hypothesis that labels all points in $\{x_1, \ldots, x_{m-1}\}$ as 1. On the other hand, if all of the $k$ examples in the compression set are dummies, then $\mathcal{A}$ outputs the hypothesis that $\tilde{A}$ outputs (restricted to $\{x_1, \ldots, x_{m-1}\}$).

We first claim that the output of $\mathcal{A}$ is $(2\varepsilon, 3\delta)$-private with respect to its input $S$.

**Claim 4.4** ($\mathcal{A}$ is private). *$\mathcal{A}$ is $(2\varepsilon, 3\delta)$-DP.*

The proof of this claim is given in Appendix A. At a high level, the privacy parameter deteriorates to $2\varepsilon$ because of the two-step process of compressing $\tilde{S}$ to $k$ points in an $\varepsilon$-DP way, and then obtaining an $\varepsilon$-DP learner thereafter.

Next, we claim that on average, there will be a lot of dummies in the compression set selected by $\tilde{A}$. This step crucially hinges on Theorem 3, where we substitute the event $E$ in the statement of the theorem to be the event that a non-dummy element is selected (i.e., $E$ is the support of the distribution $\mathcal{D}$). In particular, we get that the expected number of non-dummy elements is at most $kpe^\varepsilon + \delta kn$, which is at most $\frac{1}{32}$, if we set $p = \frac{1}{64k\varepsilon^2}$, and use that $k \leq n, \delta \leq \frac{1}{64n^2}$.

We can now reason about the error and failure probability parameters of $\mathcal{A}$. Because $\tilde{A}$ is an $(\varepsilon, \delta)$ sample DP-compression scheme that successfully learns $\mathcal{H}_m$ with error $\alpha$ and failure probability $\frac{1}{32}$, with probability at least $1 - \frac{1}{32}$ over the draw of $\tilde{S}$ and the randomness of $\tilde{A}$, the hypothesis it outputs has error at most $\alpha$. Furthermore, since the expected number of non-dummy elements chosen in the compression set is at most $\frac{1}{32}$, Markov's inequality gives that with probability at least $1 - \frac{1}{32}$ over the draw of $\tilde{S}$ and the randomness of $\tilde{A}$, all the $k$ examples chosen by $\tilde{A}$ in the compression set are dummies. By a union bound, with probability at least $1 - \frac{1}{16}$ over the draw of $\tilde{S}$ and the randomness of $\tilde{A}$, all the examples chosen to be in the compression set by $\tilde{A}$ are dummies *and* the hypothesis it outputs has error (with respect to $\tilde{\mathcal{D}}$) less than $\alpha$.

But recall that the distribution $\tilde{\mathcal{D}}$ on $\tilde{S}$ is induced by the distribution $\mathcal{D}$ on $S$, and that whenever all the examples chosen by $\tilde{A}$ in the compression set are dummies, $\mathcal{A}$ returns $\tilde{A}$'s output. This implies that with probability at least $1 - \frac{1}{16}$ over the draw of $S$ from $\mathcal{D}^n$ and the randomness of $\mathcal{A}$, the hypothesis output by $\mathcal{A}$ has error at most $\alpha$ with respect to $\tilde{\mathcal{D}}$. But since $\tilde{\mathcal{D}}$ is a mixture distribution,

$$R_{\tilde{\mathcal{D}}}(\mathcal{A}(S)) \geq p \cdot R_{\mathcal{D}}(\mathcal{A}(S)),$$

and hence we have that with probability at least $1 - \frac{1}{16}$, the error of $\mathcal{A}(S)$ with respect to $\mathcal{D}$ is at most $\frac{\alpha}{p} \leq 64ke^\varepsilon\alpha$. Thus, $\mathcal{A}$ is a $\left(64ke^\varepsilon\alpha, \frac{1}{16}, 2\varepsilon, 3\delta\right)$-DP learner for $\mathcal{H}_{m-1}$ as required. ∎

We next state a lower bound on the sample complexity of DP learners for thresholds [2, 10].

**Theorem 4** (Theorem 1 in [2]). *Let $\mathcal{H}_m$ be the class of thresholds on $\{x_1, \ldots, x_m\}$. Let $\mathcal{A}$ be a $\left(\frac{1}{16}, \frac{1}{16}, 0.1, \frac{1}{1000n^2 \log n}\right)$-DP learner for $\mathcal{H}_m$ with sample complexity n. Then $n \geq \Omega(\log^* m)$.*

We are now ready prove Theorem 5, which shows that non-Littlestone [16] classes cannot be learnt by sublinear sample DP-compression schemes. Theorem 2 follows from Theorem 5, since classes that are DP-learnable are exactly the classes with finite Littlestone dimension [11].

**Theorem 5.** *Let $\mathcal{H}$ be a hypothesis class over $\mathcal{X}$ that has infinite Littlestone dimension. For $\varepsilon = 0.05, \delta = \frac{1}{3000n^2 \log n}$, let $\tilde{\mathcal{A}}$ be an $(\varepsilon, \delta)$ sample differentially private $(n \to k)$ compression scheme that learns $\mathcal{H}$ with error $\alpha$ and failure probability $\frac{1}{32}$. Then $k \geq \frac{1}{68\alpha}$.*

*Proof.* Because $\mathcal{H}$ has infinite Littlestone dimension, for any $m \geq 1$, there exist $\{x_1, \ldots, x_m\}$ and $\mathcal{H}_m \subseteq \mathcal{H}$ such that $\mathcal{H}_m$ is the class of thresholds over $\{x_1, \ldots, x_m\}$ [2, Theorem 3]. Now, $\tilde{\mathcal{A}}$ is an $(n \to k)$ sample DP-compression scheme that learns $\mathcal{H}$; in particular, this means that $\tilde{\mathcal{A}}$ has sample complexity $n < \infty$, and also that $\tilde{\mathcal{A}}$ learns $\mathcal{H}_m$ with the same parameters and sample complexity. By Lemma 4.3, we know that there then exists a $\left(68k\alpha, \frac{1}{16}, 0.1, \frac{1}{1000n^2 \log n}\right)$ private learner for $\mathcal{H}_m$ with sample complexity $n$. Assume for the sake of contradiction that $k < \frac{1}{68\alpha}$. This means that there exists a $\left(\alpha, \frac{1}{16}, 0.1, \frac{1}{1000n^2 \log n}\right)$ private learner for $\mathcal{H}_m$ with sample complexity $n$. By Theorem 4, it must be that $n \geq \Omega(\log^* m)$. Since we can find $\mathcal{H}_m \subseteq \mathcal{H}$ for any $m \geq 1$, this would mean that $n = \infty$, which is a contradiction. Thus, it must be the case that $k \leq \frac{1}{68\alpha}$. ∎

### 4.3 Bounded boosting of empirical measure

We prove a simplified form of Theorem 3 (with slightly tighter bounds), where we consider the input to be a bit string. Theorem 3 as stated in terms of a general event can be immediately obtained as a corollary by interpreting the bits in the string as indicators for the event (details in Appendix A).

**Lemma 4.5** (Bounded Boosting of Empirical Measure). *Let $\mathcal{A} : \{0, 1\}^n \to [n]^k$ be an $(\varepsilon, \delta)$-DP selection mechanism. Let $\mathcal{D}$ be the product distribution on $\{0, 1\}^n$ where each bit is set to $1$ with probability $p$. For $X \sim \mathcal{D}$, let $Z$ denote the fraction of indices in $\mathcal{A}(X)$ at which $X$ is $1$, i.e., $Z = \frac{1}{k} \sum_{j \in \mathcal{A}(X)} \mathbb{1}[X_j = 1]$. Then, we have that*

$$\frac{p - np(1-p)\delta}{p + (1-p)e^\varepsilon} \leq \mathbb{E}[Z] \leq \frac{pe^\varepsilon + np(1-p)\delta}{1 - p + pe^\varepsilon}. \tag{4}$$

*Proof Sketch.* Let $\mathcal{A}(X) = I = (I_1, I_2, \ldots, I_k)$ be the tuple of indices selected by the DP mechanism on input $X$. We first write $Z = \frac{1}{k} \sum_{j=1}^k \sum_{i=1}^n \mathbb{1}[I_j = i \wedge X_i = 1]$. Thereafter, the main step of the proof uses that the mechanism is private in order to relate the *conditional* probability $\Pr[I_j = i | X_i = 1]$ to $\Pr[I_j = i | X_i = 0]$ for any $j \in [k]$. Concretely, observe that

$$\Pr[I_j = i | X_i = 0] = \frac{\Pr[X_i = 0 \wedge I_j = i]}{\Pr[X_i = 0]} = \frac{\sum_{x \in \{0,1\}^n, x_i = 0} \Pr[x] \Pr[I_j = i | x]}{1 - p}$$

$$= \frac{\sum_{x \in \{0,1\}^n, x_i = 1} \Pr[x^{\otimes i}] \Pr[I_j = i | x^{\otimes i}]}{1 - p} \leq \frac{\sum_{x \in \{0,1\}^n, x_i = 1} \Pr[x] \cdot (e^\varepsilon \Pr[I_j = i | x] + \delta)}{p}$$

$$= \frac{e^\varepsilon \sum_{x \in \{0,1\}^n, x_i = 1} \Pr[x] \Pr[I_j = i | x]}{p} + \delta = e^\varepsilon \cdot \Pr[I_j = i | X_i = 1] + \delta,$$

where in the fourth inequality, we used that for $x$ having $x_i = 1$, $\Pr_{\mathcal{D}}[x^{\otimes i}] = \frac{1-p}{p} \cdot \Pr[x]$, and that $\mathcal{A}$ is an $(\varepsilon, \delta)$-DP mechanism. This relation lets us express the joint probability term $\Pr[I_j = i \wedge X_i = 1]$ in the expression $\mathbb{E}[Z] = \frac{1}{k} \sum_{j=1}^k \sum_{i=1}^n \Pr[I_j = i \wedge X_i = 1]$ simply in terms of $\Pr[I_j = i]$. Thereafter, noticing that $\sum_{i=1}^n \Pr[I_j = i] = 1$ yields the result. The complete details are provided in Appendix A. ∎

## 5 Conclusion

We study two natural definitions for algorithms satisfying credit attribution. In the context of PAC learning, we provide a characterization of learnability for algorithms that respect these definitions. Our work motivates the further study of these and other related definitions for credit attribution,

and opens up interesting technical directions to pursue. However, as mentioned earlier, credit attribution is only part of the much more nuanced problem of copyright protection, and hence, our definitions only capture subtleties involved in the problem in part. With further exploration, and other suitable definitions, we will hopefully be able to ensure that algorithms (especially generative models) appropriately credit the work that they draw upon.

**Acknowledgements** Shay Moran is a Robert J. Shillman Fellow; he acknowledges support by ISF grant 1225/20, by BSF grant 2018385, by an Azrieli Faculty Fellowship, by Israel PBC-VATAT, by the Technion Center for Machine Learning and Intelligent Systems (MLIS), and by the the European Union (ERC, GENERALIZATION, 101039692). Roi Livni is supported by an ERC grant (FOG, 101116258), as well as an ISF Grant (2188 \ 20). Chirag Pabbaraju is supported by Moses Charikar and Gregory Valiant's Simons Investigator Awards. Work of Kobbi Nissim was supported by NSF Grant No. CCF2217678 "DASS: Co-design of law and computer science for privacy in sociotechnical software systems" and a gift to Georgetown University.

Views and opinions expressed are however those of the author(s) only and do not necessarily reflect those of the European Union or the European Research Council Executive Agency. Neither the European Union nor the granting authority can be held responsible for them.

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

## A Supplementary Proofs

*Proof of Claim 4.4.* Consider any 2 neighboring datasets $S = (z_1, \ldots, z_i, \ldots, z_n)$ and $S' = (z_1, \ldots, z'_i, \ldots, z_n)$ that differ at index $i$. Here, we are using the shorthand $z_i = (x_i, y_i)$. We want to argue that the distribution of $\mathcal{A}(S) =_{2\varepsilon, 3\delta} \mathcal{A}(S')$. Let $O$ be any subset of the output space of $\mathcal{A}$. Recall that $\mathcal{A}$ first constructs the sample $\tilde{S}$ from $S$ and then passes it to the semi-private learner $\tilde{\mathcal{A}}$. Then,

$$\Pr[\mathcal{A}(S) \in O] = \Pr[\mathcal{A}(S) \in O|z_i \in \tilde{S}] \Pr[z_i \in \tilde{S}] + \Pr[\mathcal{A}(S) \in O|z_i \notin \tilde{S}] \Pr[z_i \notin \tilde{S}]$$

$$= \Pr[\mathcal{A}(S) \in O|z_i \in \tilde{S}] \Pr[z'_i \in \tilde{S}'] + \Pr[\mathcal{A}(S') \in O|z'_i \notin \tilde{S}'] \Pr[z'_i \notin \tilde{S}'] \quad (5)$$

where we used that the coins that deterine whether $z_i \in S$ (or $z'_i \in \tilde{S}'$) are tossed independently of the data, and that the distribution of $\tilde{S}'$ conditioned on $z'_i \notin \tilde{S}'$, is identical to the distribution of $\tilde{S}$ conditioned on $z_i \notin \tilde{S}$. Hence, we focus on the term $\Pr[\mathcal{A}(S) \in O|z_i \in \tilde{S}]$ in (5). We can decompose this as

$$\Pr[\mathcal{A}(S) \in O|z_i \in \tilde{S}] = \sum_{\tilde{s}:z_i \in \tilde{s}} \Pr[\mathcal{A}(S) \in O|z_i \in \tilde{S}, \tilde{S} = \tilde{s}] \Pr[\tilde{S} = \tilde{s}|z_i \in \tilde{S}]$$

$$= \sum_{\tilde{s}':z'_i \in \tilde{s}'} \Pr[\mathcal{A}(S) \in O|z_i \in \tilde{S}, \tilde{S} = \tilde{s}] \Pr[\tilde{S}' = \tilde{s}'|z'_i \in \tilde{S}']. \quad (6)$$

Here, for every term in the summation, $\tilde{s}'$ differs from $\tilde{s}$ at exactly one index $i$, and we again used that the coins used to construct $\tilde{S}$ and $\tilde{S}'$ are independent of the data. Let $E(\tilde{s})$ be the event that all the $k$ samples chosen by the semi-private learner $\tilde{\mathcal{A}}$ when it is given $\tilde{s}$ as input are dummies. Since $\tilde{s}$ and $\tilde{s}'$ differ in exactly one element, because of the special property of the selection mechanism of $\tilde{\mathcal{A}}$, we have that

$$\Pr[E(\tilde{s})|z_i \in \tilde{S}, \tilde{S} = \tilde{s}] \leq e^\varepsilon \cdot \Pr[E(\tilde{s}')|z'_i \in \tilde{S}', \tilde{S}' = \tilde{s}'] + \delta \quad (7)$$

$$\Pr[\neg E(\tilde{s})|z_i \in \tilde{S}, \tilde{S} = \tilde{s}] \leq e^\varepsilon \cdot \Pr[\neg E(\tilde{s}')|z'_i \in \tilde{S}', \tilde{S}' = \tilde{s}'] + \delta. \quad (8)$$

But note that the set of public examples is exactly the same, if $E(\tilde{s})$ and $E(\tilde{s}')$ respectively occur—hence, the learner in $\tilde{\mathcal{A}}$ (which is a function of the set of public examples) that operates on the private examples in either case is identical. Furthermore, the sets of private examples themselves differ in exactly one element; we can thus use the privacy guarantees of the learner in $\tilde{\mathcal{A}}$ to claim that

$$\Pr[\mathcal{A}(S) \in O|z_i \in \tilde{S}, \tilde{S} = \tilde{s}, E(\tilde{s})] \leq \min\left(1, e^\varepsilon \cdot \Pr[\mathcal{A}(S') \in O|z'_i \in \tilde{S}', \tilde{S}' = \tilde{s}', E(\tilde{s}')]\right) + \delta. \quad (9)$$

Combining (7) and (9), we get

$$\Pr[\mathcal{A}(S) \in O|z_i \in \tilde{S}, \tilde{S} = \tilde{s}, E(\tilde{s})] \cdot \Pr[E(\tilde{s})|z_i \in \tilde{S}, \tilde{S} = \tilde{s}]$$

$$\leq \left(\min\left(1, e^\varepsilon \cdot \Pr[\mathcal{A}(S') \in O|z'_i \in \tilde{S}', \tilde{S}' = \tilde{s}', E(\tilde{s}')]\right) + \delta\right) \Pr[E(\tilde{s})|z_i \in \tilde{S}, \tilde{S} = \tilde{s}]$$

$$\leq \min\left(1, e^\varepsilon \cdot \Pr[\mathcal{A}(S') \in O|z'_i \in \tilde{S}', \tilde{S}' = \tilde{s}', E(\tilde{s}')]\right) \Pr[E(\tilde{s})|z_i \in \tilde{S}, \tilde{S} = \tilde{s}] + \delta$$

$$\leq \min\left(1, e^\varepsilon \cdot \Pr[\mathcal{A}(S') \in O|z'_i \in \tilde{S}', \tilde{S}' = \tilde{s}', E(\tilde{s}')]\right) \left(e^\varepsilon \cdot \Pr[E(\tilde{s}')|z'_i \in \tilde{S}', \tilde{S}' = \tilde{s}'] + \delta\right) + \delta$$

$$\leq e^{2\varepsilon} \cdot \Pr[\mathcal{A}(S') \in O|z'_i \in \tilde{S}', \tilde{S}' = \tilde{s}', E(\tilde{s}')] \cdot \Pr[E(\tilde{s}')|z'_i \in \tilde{S}', \tilde{S}' = \tilde{s}'] + 2\delta. \quad (10)$$

Now, observe that if $E(\tilde{s})$ does not occur (and correspondingly if $E(\tilde{s}')$ does not occur), then we deterministically out the constant hypothesis in either case, and hence

$$\Pr[\mathcal{A}(S) \in O|z_i \in \tilde{S}, \tilde{S} = \tilde{s}, \neg E(\tilde{s})] = \Pr[\mathcal{A}(S') \in O|z'_i \in \tilde{S}', \tilde{S}' = \tilde{s}', \neg E(\tilde{s}')]. \quad (11)$$

Combining (8) and (11), we get

$$\Pr[\mathcal{A}(S) \in O|z_i \in \tilde{S}, \tilde{S} = \tilde{s}, \neg E(\tilde{s})] \cdot \Pr[\neg E(\tilde{s})|z_i \in \tilde{S}, \tilde{S} = \tilde{s}]$$

$$\leq e^\varepsilon \cdot \Pr[\mathcal{A}(S') \in O|z'_i \in \tilde{S}', \tilde{S}' = \tilde{s}', \neg E(\tilde{s}')] \cdot \Pr[\neg E(\tilde{s}')|z'_i \in \tilde{S}', \tilde{S}' = \tilde{s}'] + \delta \quad (12)$$

Altogether, (10) and (12) give that

$$\Pr[\mathcal{A}(S) \in O|z_i \in \tilde{S}, \tilde{S} = \tilde{s}] \leq e^{2\varepsilon} \cdot \Pr[\mathcal{A}(S') \in O|z'_i \in \tilde{S}', \tilde{S}' = \tilde{s}'] + 3\delta.$$

Substituting in (6), we get

$$\Pr[\mathcal{A}(S) \in O | z_i \in \tilde{S}] \leq \sum_{\tilde{s}' : z_i' \in \tilde{s}'} \Pr[\mathcal{A}(S) \in O | z_i \in \tilde{S}, \tilde{S} = \tilde{s}] \Pr[\tilde{S}' = \tilde{s}' | z_i' \in \tilde{S}']$$

$$\leq \sum_{\tilde{s}' : z_i' \in \tilde{s}'} \left( e^{2\varepsilon} \cdot \Pr[\mathcal{A}(S') \in O | z_i' \in \tilde{S}', \tilde{S}' = \tilde{s}'] + 3\delta \right) \Pr[\tilde{S}' = \tilde{s}' | z_i' \in \tilde{S}']$$

$$\leq 3\delta + e^{2\varepsilon} \sum_{\tilde{s}' : z_i' \in \tilde{s}'} \Pr[\mathcal{A}(S') \in O | z_i' \in \tilde{S}', \tilde{S}' = \tilde{s}'] \Pr[\tilde{S}' = \tilde{s}' | z_i' \in \tilde{S}']$$

$$= e^{2\varepsilon} \cdot \Pr[\mathcal{A}(S') \in O | z_i' \in \tilde{S}'] + 3\delta.$$

Finally, substituting back in (5), we get

$$\Pr[\mathcal{A}(S) \in O]$$

$$\leq \left( e^{2\varepsilon} \cdot \Pr[\mathcal{A}(S') \in O | z_i' \in \tilde{S}'] + 3\delta \right) \Pr[z_i' \in \tilde{S}'] + \Pr[\mathcal{A}(S') \in O | z_i' \notin \tilde{S}'] \Pr[z_i' \notin \tilde{S}']$$

$$\leq e^{2\varepsilon} \cdot \left( \Pr[\mathcal{A}(S') \in O | z_i' \in \tilde{S}'] \Pr[z_i' \in \tilde{S}'] + \Pr[\mathcal{A}(S') \in O | z_i' \notin \tilde{S}'] \Pr[z_i' \notin \tilde{S}'] \right) + 3\delta$$

$$= e^{2\varepsilon} \cdot \Pr[\mathcal{A}(S') \in O] + 3\delta.$$

By the same calculations, we also get the bound $\Pr[\mathcal{A}(S') \in O] \leq e^{2\varepsilon} \cdot \Pr[\mathcal{A}(S) \in O] + 3\delta$, completing the proof. ∎

*Proof of Lemma 4.5.* Let $\mathcal{A}(X) = I = (I_1, I_2, \ldots, I_k)$. Note that

$$Z = \frac{1}{k} \sum_{j=1}^{k} \sum_{i=1}^{n} \mathbb{1}[I_j = i \wedge X_i = 1],$$

and hence

$$\mathbb{E}[Z] = \frac{1}{k} \sum_{j=1}^{k} \sum_{i=1}^{n} \Pr_{X,\mathcal{A}}[I_j = i \wedge X_i = 1] = \frac{1}{k} \sum_{j=1}^{k} \sum_{i=1}^{n} \underbrace{\Pr[X_i = 1]}_{=p} \cdot \Pr[I_j = i | X_i = 1]$$

$$= \frac{p}{k} \cdot \sum_{j=1}^{k} \sum_{i=1}^{n} \Pr[I_j = i | X_i = 1]. \tag{13}$$

Now, for any $x \in \{0, 1\}^n$, let $x^{\otimes i}$ denote $x$ with its $i^{\text{th}}$ bit flipped. Then, observe that

$$\Pr[I_j = i | X_i = 0] = \frac{\Pr[X_i = 0 \wedge I_j = i]}{\Pr[X_i = 0]} = \frac{\sum_{x \in \{0,1\}^n, x_i=0} \Pr[x] \Pr[I_j = i | x]}{1 - p}$$

$$= \frac{\sum_{x \in \{0,1\}^n, x_i=1} \Pr[x^{\otimes i}] \Pr[I_j = i | x^{\otimes i}]}{1 - p} \leq \frac{\sum_{x \in \{0,1\}^n, x_i=1} \Pr[x] \cdot (e^\varepsilon \Pr[I_j = i | x] + \delta)}{p}$$

$$= \frac{e^\varepsilon \sum_{x \in \{0,1\}^n, x_i=1} \Pr[x] \Pr[I_j = i | x]}{p} + \delta = e^\varepsilon \cdot \Pr[I_j = i | X_i = 1] + \delta,$$

where in the fourth inequality, we used that for $x$ having $x_i = 1$, $\Pr_{\mathcal{D}}[x^{\otimes i}] = \frac{1-p}{p} \cdot \Pr[x]$, and that $\mathcal{A}$ is an $(\varepsilon, \delta)$-DP mechanism. Hence, we have that

$$\Pr_{X,\mathcal{A}}[I_j = i] = \Pr[X_i = 0] \cdot \Pr[I_j = i | X_i = 0] + \Pr[X_i = 1] \cdot \Pr[I_j = i | X_i = 1]$$

$$\leq \Pr[X_i = 0] \cdot (e^\varepsilon \cdot \Pr[I_j = i | X_i = 1] + \delta) + \Pr[X_i = 1] \cdot \Pr[I_j = i | X_i = 1]$$

$$= (p + e^\varepsilon (1 - p)) \Pr[I_j = i | X_i = 1] + (1 - p)\delta \tag{14}$$

$$\implies \quad \Pr[I_j = i | X_i = 1] \geq \frac{\Pr[I_j = i] - (1 - p)\delta}{p + e^\varepsilon (1 - p)}. \tag{15}$$

Substituting (15) in (13), we get

$$\mathbb{E}[Z] \geq \frac{p}{k(p + e^\varepsilon (1 - p))} \cdot \left( \sum_{j=1}^{k} \sum_{i=1}^{n} \Pr[I_j = i] - nk(1 - p)\delta \right). \tag{16}$$

Finally, note that $\sum_{i=1}^n \Pr[I_j = i] = 1$ for any $j$. Substituting in (16), we have shown the desired lower bound

$$\mathbb{E}[Z] \geq \frac{p - np(1-p)\delta}{p + e^\varepsilon(1-p)}.$$

For the upper bound, we repeat the above analysis with $Z' = \frac{1}{k}\sum_{j \in \mathcal{A}(X)} \mathbb{1}[X_j = 0]$, to obtain

$$\mathbb{E}[Z'] \geq \frac{(1-p) - np(1-p)\delta}{1 - p + pe^\varepsilon}.$$

But note that $Z' = 1 - Z$, and hence

$$\mathbb{E}[Z] = 1 - \mathbb{E}[Z'] \leq \frac{pe^\varepsilon + np(1-p)\delta}{1 - p + pe^\varepsilon}.$$

$\blacksquare$

*Proof of Theorem 3.* Recall that $E \subseteq \mathcal{Z}$ is an event satisfying $\mathcal{D}(E) = p$ for the given distribution $\mathcal{D}$ over $\mathcal{Z}$. Let $\mathcal{D}|E$ denote the distribution $\mathcal{D}$ conditioned on the event $E$, and let $\mathcal{D}|\neg E$ denote the distribution $\mathcal{D}$ conditioned on the complement of event $E$. Assume for the sake of contradiction that either $\mathbb{E}[Z] > \frac{pe^\varepsilon+np(1-p)\delta}{1-p+pe^\varepsilon}$ or $\mathbb{E}[Z] < \frac{p+np(1-p)\delta}{p+(1-p)e^\varepsilon}$. Then, consider an algorithm $\mathcal{B}$, that takes as input a bit string $Y$ from a product distribution on $\{0,1\}^n$, where each bit is independently set to 1 with probability $p$. Given such an input string $Y$, the algorithm constructs a sequence $S = \{z_1, \ldots, z_n\}$, where $z_i \sim \mathcal{D}|E$ if $Y_i = 1$, and $z_i \sim \mathcal{D}|\neg E$ otherwise. Thus, $S$ is exactly distributed as $D^n$. $\mathcal{B}$ then passes $S$ to the DP sample compression scheme $M$, which selects a compression set $\kappa(S) = (i_1, \ldots, i_k)$—this is the tuple of indices that $\mathcal{B}$ outputs too. Note that because the compression function $\kappa$ is an $(\varepsilon, \delta)$-DP mechanism, $\mathcal{B}$ is also an $(\varepsilon, \delta)$-DP mechanism with respect to its input. To see this, consider two neighboring bit strings $y, y'$, such that $y_i = 1$ and $y'_i = 0$. We will show that $\Pr[\mathcal{B}(y) \in O] \leq e^\varepsilon \cdot \Pr[\mathcal{B}(y') \in O] + \delta$, and the same calculations will give the bound with $y, y'$ swapped.

$$\Pr[\mathcal{B}(y) \in O] = \sum_{z_{-i}} \Pr[z_{-i}] \sum_{z_i \in E} \Pr[z_i|E] \Pr[\mathcal{A}(z_{-i} \circ z_i) \in O] \tag{17}$$

Now, for any $z'_i \in \neg E$, we know (since $\kappa$ is an $(\varepsilon, \delta)$-DP mechanism) that

$$\Pr[\mathcal{A}(z_{-i} \circ z_i) \in O] \leq e^\varepsilon \cdot \Pr[\mathcal{A}(z_{-i} \circ z'_i) \in O] + \delta,$$

and hence

$$\Pr[\mathcal{A}(z_{-i} \circ z_i) \in O] \leq e^\varepsilon \cdot \sum_{z'_i \in \neg E} \Pr[z'_i|\neg E] \Pr[\mathcal{A}(z_{-i} \circ z'_i) \in O] + \delta. \tag{18}$$

Substituting (18) in (17) gives that

$$\Pr[\mathcal{B}(y) \in O] \leq e^\varepsilon \cdot \sum_{z_{-i}} \Pr[z_{-i}] \sum_{z'_i \in \neg E} \Pr[z'_i|\neg E] \Pr[\mathcal{A}(z_{-i} \circ z'_i) \in O] + \delta$$

$$= e^\varepsilon \cdot \Pr[\mathcal{B}(y') \in O] + \delta.$$

Now, by our assumption, either $\mathbb{E}[Z] > \frac{pe^\varepsilon+np(1-p)\delta}{1-p+pe^\varepsilon}$ or $\mathbb{E}[Z] < \frac{p+np(1-p)\delta}{p+(1-p)e^\varepsilon}$. But this means that either $\mathbb{E}\left[\sum_{j=1}^k \mathbb{1}[Y_{i_j} = 1]\right] > \frac{pe^\varepsilon+np(1-p)\delta}{1-p+pe^\varepsilon}$ or $\mathbb{E}\left[\sum_{j=1}^k \mathbb{1}[Y_{i_j} = 1]\right] < \frac{p+np(1-p)\delta}{p+(1-p)e^\varepsilon}$. Thus, $\mathcal{B}$ is an $(\varepsilon, \delta)$-DP selection mechanism that violates the bounds in Lemma 4.5, and hence our assumption is false. $\blacksquare$

## B  A DP sample compression scheme based on Randomized Response

**Definition 7** (Randomized response). *Let* $\mathrm{RR} : \{0,1\}^n \to [n]^k$ *be the randomized response selection mechanism defined as follows. Given* $x \in \{0,1\}^n$, $\mathrm{RR}$ *flips each bit of* $x$ *independently with probability* $\frac{1}{1+e^\varepsilon}$ *to obtain* $\tilde{x}$. *Let* $S = \{i \in [n] : \tilde{x}_i = 1\}$ *and* $S' = [n] \setminus S$. *Further, let* $|S| = t$. *If* $t \geq k$, *then* $\mathrm{RR}$ *outputs a uniformly random subset of* $k$ *indices from* $S$, *ordered arbitrarily. Otherwise, it arbitrarily orders* $S$, *and outputs* $S \circ T$, *where* $T$ *is a uniformly random subset of* $k - t$ *indices chosen from* $S'$ *(and ordered arbitrarily), and* $\circ$ *denotes concatenation.*

**Claim B.1** (RR boosts empirical measure optimally). *In the setting of Lemma 4.5, let $\delta = 0$ and let $\mathcal{A}$ be the randomized response mechanism RR (Definition 7). Then,*

$$\mathbb{E}[Z] \geq \left(1 - kn^k \cdot \exp\left(\frac{(k-n)(1-p+pe^\varepsilon)}{1+e^\varepsilon}\right)\right) \cdot \frac{pe^\varepsilon}{1-p+pe^\varepsilon}. \tag{19}$$

**Remark 3.** Observe that when $k = o\left(\frac{n}{\log n}\right)$ and $n$ gets large, the expression in the parentheses approaches 1. Thus, we can conclude that randomized response attains the upper bound from Lemma 4.5 when $\delta = 0$.

*Proof.* Recall that for $X \sim \mathcal{D}$, randomized response first constructs $Y$ by flipping each bit of $X$ with probability $\frac{1}{1+e^\varepsilon}$. That is, the distribution of $Y$ is the product distribution where each $Y_i$ is 1 with probability $p \cdot \frac{e^\varepsilon}{1+e^\varepsilon} + (1-p) \cdot \frac{1}{1+e^\varepsilon} = \frac{1-p+pe^\varepsilon}{1+e^\varepsilon} := \alpha$. Let $S = \{i \in [n] : Y_i = 1\}$. We first claim that with high probability, $|S| \geq k$.

To see this, note that

$$\begin{aligned}
\Pr[|S| \geq k] &= 1 - \Pr[|S| < k] \\
&= 1 - \Pr[\exists S' \in [n] : |S'| > n - k, Y_i = 0 \; \forall i \in S'] \\
&\geq 1 - \Pr[\exists S' \in [n] : |S'| = n - k + 1, Y_i = 0 \; \forall i \in S'] \\
&\geq 1 - \binom{n}{k-1}(1-\alpha)^{n-k+1} \\
&\geq 1 - \underbrace{n^k \cdot e^{-\alpha(n-k)}}_{:=\beta},
\end{aligned}$$

where we denote the tail probability by $\beta$. Note that since we assume $k = o\left(\frac{n}{\log n}\right)$, $\beta = o(1)$.

Let $I$ be the tuple of $k$ indices that RR outputs (note that all these indices are always distinct). Recall that, if $|S| \geq k$, then $Y_i = 1$ for all $i \in I$. Let $Z_1 = \sum_{i \in I} \mathbb{1}[Y_i = 1]$ and $Z_0 = \sum_{i \in I} \mathbb{1}[Y_i = 0]$. Then we have that

$$\begin{aligned}
\mathbb{E}[Z_1] &\geq k \cdot \Pr[|S| \geq k] \geq k(1-\beta) \\
\implies \quad \mathbb{E}[Z_0] &\leq k\beta.
\end{aligned}$$

Markov's inequality then gives us that $\Pr[Z_0 \geq 1] \leq k\beta$. Thus, with probability at least $1 - k\beta$, we have that $Y_i = 1$ for all (distinct) indices output by RR: let $G$ denote this event.

Finally, let $Z = \frac{1}{k}\sum_{i \in I} \mathbb{1}[X_i = 1]$. Then, we have that

$$\begin{aligned}
\mathbb{E}[Z] &\geq \Pr[G] \cdot \mathbb{E}[Z|G] \\
&\geq (1-k\beta) \cdot \mathbb{E}[Z|G] \\
&= (1-k\beta) \cdot \sum_{i_1,\ldots,i_k} \Pr[I = \{i_1,\ldots,i_k\}|G] \cdot \mathbb{E}[Z|G, I = \{i_1,\ldots,i_k\}]. \tag{20}
\end{aligned}$$

But observe that

$$\begin{aligned}
\mathbb{E}[Z|G, I = \{i_1,\ldots,i_k\}] &= \frac{1}{k}\sum_{j \in \{i_1,\ldots,i_k\}} \Pr[X_j = 1|Y_j = 1] \\
&= \frac{1}{k}\sum_{j \in \{i_1,\ldots,i_k\}} \frac{\Pr[X_j = 1 \wedge Y_j = 1]}{\Pr[Y_j = 1]} \\
&= \frac{1}{k}\sum_{j \in \{i_1,\ldots,i_k\}} \frac{p \cdot \frac{e^\varepsilon}{1+e^\varepsilon}}{p \cdot \frac{e^\varepsilon}{1+e^\varepsilon} + (1-p) \cdot \frac{1}{1+e^\varepsilon}} \\
&= \frac{pe^\varepsilon}{1-p+pe^\varepsilon}.
\end{aligned}$$

Substituting in (20) and plugging in the expression for $\beta$, we get the desired bound

$$\mathbb{E}[Z] \geq \left(1 - kn^k \cdot \exp\left(\frac{(k-n)(1-p+pe^\varepsilon)}{1+e^\varepsilon}\right)\right) \cdot \frac{pe^\varepsilon}{1-p+pe^\varepsilon}.$$

∎

