# OpenReview forum: "Credit Attribution and Stable Compression"
_NeurIPS.cc/2024/Conference — NeurIPS 2024 poster_

### Official Review · Reviewer_ariQ · 2024-06-28

**Soundness:** 3
**Presentation:** 3
**Contribution:** 2
**Rating:** 5
**Confidence:** 3

**Summary:**

This paper studies credit attribution and stable compression. Credit attribution aims to assign recognition to the original creators of content generated by machine learning models. The authors propose formal definitions for credit attribution and connect it to differential privacy (DP). The proposed framework extends notions of stability, such as stable sample compression. The paper uses the PAC learning framework to study the learnability of machine-learning problems under the constraints of counterfactual credit attribution (CCA) and stable sample compression.

**Strengths:**

1. The paper proposes formal definitions for credit attribution.
2. It shows that every PAC learnable class can also be learned using a CCA learning rule.
3. It connects CCA and sample compression to differential privacy.
4. It characterizes the expressive power of different learning rules, such as CCA, sample compression, and differentially private learning, using the PAC learning framework.

**Weaknesses:**

1. The paper motivates the contributions using generative models but presents results only for PAC learnable classes. Showing how credit attribution could be applied to generative models would enhance the paper.
2. Even for classification problems, the paper does not provide examples of credit attribution on common tasks such as image and text classification. Some examples of applications of CCA would significantly improve the impact of this work.
3. The paper could benefit from connecting its contributions to existing literature. Adding a section on related work and positioning the contributions within the broader literature would provide valuable context.
4. The intuition behind semi-differentially private learning is not very clear. An explanation using a concrete example would be helpful.

Minor Correction: Sec 2, Line 71: X* -> Z*

**Questions:**

1. How could this framework be extended to generative models, such as text and image generators?
2. How can the quality of the credit attributions be evaluated? Is there a risk that a credit attribution mechanism could mistakenly assign credit to creators who are unrelated to the generated content (along with the actual creators on which the generated content was based)?

**Limitations:**

Yes, the authors have addressed the limitations of their work at different points in the paper, but not in a separate "Limitations" section, as encouraged in the guidelines.

---

> ### Author Rebuttal · Authors · 2024-08-07
>
> Thank you so much for reading our paper, and for your comments.
>
> With regards to the weaknesses you bring up:
> > 1. The paper motivates the contributions using generative models but presents results only for PAC learnable classes. Showing how credit attribution could be applied to generative models would enhance the paper.
> 2. Even for classification problems, the paper does not provide examples of credit attribution on common tasks such as image and text classification. Some examples of applications of CCA would significantly improve the impact of this work.
>
> Our definitions for credit attribution (CCA and DP sample compression schemes) apply verbatim to any mechanisms in general, including generative models producing text/images. As such, they merely formalize the requirement that a mechanism satisfy the principle of counterfactual attribution.
>
> > 3. The paper could benefit from connecting its contributions to existing literature. Adding a section on related work and positioning the contributions within the broader literature would provide valuable context.
>
> We have elaborated on the many connections of our work (along with ample references) to the existing literature on differential privacy, stable compression schemes, private learning with public data, as well as copyright management throughout the Introduction and Definitions sections (Sections 1 and 2). If there are concrete examples of references that we missed, please let us know and we will be glad to incorporate those in the final manuscript.
>
> > 4. The intuition behind semi-differentially private learning is not very clear. An explanation using a concrete example would be helpful.
>
> Semi-differentially private learning is applicable in settings where it is conceivable to have access to a corpus of public data, in addition to some sensitive, private data. For example, many public surveys collecting data of some sort routinely have an option for users to “opt-in” to voluntarily allow the organization to use their data in a non-confidential manner. Users who do not care too much about the privacy of their data could choose to opt-in, while others that are more worried about privacy could choose not to. We can also imagine settings where a large corpus of unlabeled, public data is freely available (e.g., unlabeled images available online), but obtaining labeled data is expensive and necessitates preserving privacy. In such cases, where we have access to public data on which privacy constraints can be relaxed, it is reasonable to ask if we can design mechanisms that have better performance guarantees compared to the setting requiring full privacy. And as it turns out, this is indeed true—while simple function classes like thresholds on the unit interval cannot be efficiently learned under complete differential privacy, this task becomes much more efficient under semi-differential privacy. Further background about semi-differential privacy is provided in the works [1,2,3].
>
> [1] Amos Beimel, Kobbi Nissim, and Uri Stemmer. Private learning and sanitization: Pure vs. approximate differential privacy.
>
> [2] Amos Beimel, Kobbi Nissim, and Uri Stemmer. Learning privately with labeled and unlabeled examples.
>
> [3] Noga Alon, Raef Bassily, and Shay Moran. Limits of private learning with access to public data.
>
> ---
> With regards to your questions:
>
> > 1. How could this framework be extended to generative models, such as text and image generators?
>
> Thanks for this question, we should indeed add a discussion on the challenges in establishing for which classes generative learning is possible within our setup. In DP, there is a known close relationship between classes that are PAC learnable and classes that one can produce private synthetic data. There are certain challenges in proving a similar connection between PAC learning and synthetic data generation in our framework, and we believe this is an important future work. But overall, the first key step in understanding which classes one can generate synthetic data for is understanding what is PAC learnable.
>
> > 2. How can the quality of the credit attributions be evaluated? Is there a risk that a credit attribution mechanism could mistakenly assign credit to creators who are unrelated to the generated content (along with the actual creators on which the generated content was based)?
>
> Mechanisms satisfying our definitions are technically allowed to cite superfluous and possibly unrelated works; however, this is similar in spirit to the “Precision vs Recall” tradeoff. In the context of credit attribution, our definitions focus primarily on the “Recall” aspect of the problem—namely, an algorithm should not miss out on citing any work if its output derives heavily from it, even if it cites some extraneous works. To us, this seems like the more pressing objective of the two, owing to legality concerns: the owner of the work that the algorithm failed to acknowledge could sue in court. On the other hand, while the “Precision” problem of citing needless other works does seem like an issue, it appears to have less drastic implications. It is an interesting direction to further add the precision constraint in our definitions.
>
> ---
> Please let us know if we can answer any more questions!

---

> > ### Comment · Reviewer_nyTY · 2024-08-10
> >
> > Thank you for the clarification. I would change my rating and suggest acceptance of the paper.

---

> > > ### Author Response · Authors · 2024-08-11
> > > **Thanks**
> > >
> > > Thank you! Just a quick reminder to raise the score when you can.

---

> > > ### Comment · Reviewer_ariQ · 2024-08-12
> > >
> > > Dear Reviewer nyTY,
> > >
> > > It seems that this comment may have been posted under the wrong rebuttal. Could you please review and respond to the correct one?
> > >
> > > Thank you for your attention to this matter.
> > >
> > > Best regards,
> > > Reviewer ariQ

---

> > ### Comment · Reviewer_ariQ · 2024-08-12
> >
> > I would like to thank the authors for their responses to my comments and questions. While some of my concerns have been addressed, my main concerns still remain.
> >
> > 1. The proposed framework for credit attribution does not seem directly applicable to the generative models discussed in the abstract and introduction as a motivation for this work. If the framework is indeed applicable to such models, it would be beneficial to include a clear example demonstrating this. If not, it might be advisable to reduce the emphasis on generative models in the early sections of the paper. Otherwise, this emphasis could potentially mislead readers into believing that the framework is effective for credit attribution in the context of generative models.
> > 2. As I understand it, the paper examines the proposed framework for support vector machines in the context of classification. While this contribution is valuable, it would be even more impactful to demonstrate credit attribution for modern machine learning models, such as convolutional neural networks used for image classification. If the framework does not apply to such models, it would be helpful to include a comparison between the performance of state-of-the-art classification models and those that support credit attribution. This would provide valuable insights into the performance trade-offs associated with incorporating credit attribution into classification tasks.
> > 3. While it is understandable to prioritize recall in the context of credit attribution, it is equally important to ensure that precision remains at a reasonable level. If precision is allowed to be too low, it would be trivial to achieve perfect recall simply by attributing credit to all training samples. An empirical analysis of the trade-off between recall and precision within the proposed framework would be highly beneficial.

---

> > > ### Author Response · Authors · 2024-08-13
> > > **Response to comment by Reviwer ariQ**
> > >
> > > Thank you again for your response. We really appreciate your time in engaging in this discussion!
> > > 1) We would like to clarify again that our definitions are valid for any mechanisms in general, **including generative models**. More explicitly, we can think of a generative model as a CCA mechanism $M: Z^*\to C \times Z^*$ satisfying Definition 1, where the input $Z^*$ is the training data that the model sees (e.g., existing artworks), and the output $C$ is the new content it generates (e.g., new artwork) and $Z^*$ is the artworks that it cites. In this sense, the definition is general and *is* effective for credit attribution in the context of generative models (it is just a criterion that the model must satisfy). Perhaps it also helps to keep in mind how Differential Privacy is a definition applicable to mechanisms in general, has a well-established theory in the PAC learning setting, and at the same time is also applied in practice for many other types of algorithms. Please let us know of an example if there is something more specific that you are looking for!
> > > 2) SVM is just an example of a learning algorithm that (conveniently) already satisfies the definition that we propose for credit attribution. In general, modern ML algorithms like CNNs and Transformers, by themselves do not satisfy our definitions. It is also well-agreed at this point that these models significantly outperform SVMs on benchmark tasks (e.g., image classification). One of our motivations behind proposing the definitions that we do (which, again are general criterions) is that the community can start to modify and suitably adapt modern ML algorithms like CNNs so that they adhere to the credit attribution criterion.
> > > 3) We agree that it is always trivially possible to get full recall, by just citing the entire training dataset. However, the objective is to cite only a small-sized, relevant portion of the training data. In this sense, our PAC learning algorithm that satisfies CCA (Theorem 1) obtains meaningful bounds---even in the **worst case**, it cites only $k=O(d\log{n})$ examples, upon seeing a training dataset of $n$ examples labeled by a hypothesis class of VC dimension $d$. As we state, it would be an interesting future direction to obtain optimal, distribution dependent bounds that possibly also factor in precision.

---

### Official Review · Reviewer_nyTY · 2024-07-01

**Soundness:** 3
**Presentation:** 2
**Contribution:** 3
**Rating:** 6
**Confidence:** 3

**Summary:**

The authors propose a new notion of credit attribution, which is a relaxation of differential privacy (DP). They discuss the relationship between the proposed notion, semi-differential privacy where part of the data records are public, and a DP sample compression scheme. PAC learning theory under these notions is investigated.

**Strengths:**

The proposed credit attribution notion is interesting, and the established relationship between this notion, semi-differential privacy, and stable sample compression is intriguing. The PAC learning part seems correct, and the results intuitively make sense, even though I haven't read the detailed proof.

**Weaknesses:**

Even though the proposed new definition of credit attribution may provide a new perspective on copyright and privacy, the current content may not support this strongly due to the following weaknesses:

1. In the introduction, the authors discussed the applications of credit attribution, particularly in copyright analysis. However, in the rest of the paper, they did not mention any applications of the proposed credit attribution notion (Definition 1). In fact, I am quite curious about the connection between Definition 1 and copyright analysis (such as in Section 2 of "On Provable Copyright Protection for Generative Models").


To me, it is necessary to establish a solid mathematical connection between Definition 1 (or Definition 3) and existing widely used copyright notions to convince readers that the proposed notion is truly useful. If not, an empirical study of the proposed Definition 1 or 3 in real-world algorithms, such as how to achieve the proposed definition by modifying SGD (just like DP-SGD, which is a DP counterpart of SGD), is necessary.

2. The authors claimed that Definition 1 extends the semi-DP notion in Definition 2. However, this extension is not clearly stated in the current version. Specifically, I was confused about how to convert a special case of Definition 1 to Definition 2 (which part is public in Definition 1). Moreover, why is Definition 2 a stronger notion than Definition 1, as claimed by the authors?

It would be a significant enhancement if the authors could establish a solid mathematical relationship between these three definitions.

**Questions:**

Please clarify my concerns in the weaknesses section. Additionally, please point out if I made any mistakes in the review.

I believe this paper has the potential to be accepted as it might provide a new perspective on copyright and privacy analysis. However, for the current version, I would not suggest accepting it due to the weaknesses I listed.

**Limitations:**

Yes

---

> ### Author Rebuttal · Authors · 2024-08-07
>
> Thank you very much for reviewing our manuscript. We appreciate your feedback and would like to clarify the scope and objectives of our paper to facilitate a more accurate assessment. The paper aims to explore and propose notions of credit attribution. It does not cover topics such as copyright analysis or the study of SGD.
>
> We now address your concerns:
>
> > In the introduction, the authors discussed the applications of credit attribution, particularly in copyright analysis. However, in the rest of the paper, they did not mention any applications of the proposed credit attribution notion (Definition 1). In fact, I am quite curious about the connection between Definition 1 and copyright analysis (such as in Section 2 of "On Provable Copyright Protection for Generative Models").
>
> Please notice that, while we see the problem of credit attribution as **part** of the larger problem of copyright, we explicitly state that the paper **does not deal with the copyright question as a whole but only with a particular aspect.** Therefore, providing any copyright analysis is not within the scope of this paper, and we ask that the paper be judged by its merits and not by objectives that the paper doesn’t presume to tackle.
>
> > To me, it is necessary to establish a solid mathematical connection between Definition 1 (or Definition 3) and existing widely used copyright notions to convince readers that the proposed notion is truly useful. If not, an empirical study of the proposed Definition 1 or 3 in real-world algorithms, such as how to achieve the proposed definition by modifying SGD (just like DP-SGD, which is a DP counterpart of SGD), is necessary.
>
> We are also explicit that the notion of credit attribution that we consider is essentially **orthogonal** to previous work and existing copyright notions –-- while their focus is on the question of *substantial similarity*, we focus on algorithms that are allowed to be influenced by previous work (and may even be substantially similar), but must attribute credit. Summarily, these are two different tasks.
>
> As to the analysis of SGD, in this paper we suggest a general framework for credit attribution, and as such, we intentionally do not focus on a specific algorithm or a specific use-case of the notion. While designing an SGD version for our algorithm may be an interesting future work, it is not within the scope of the current paper.
>
> > The authors claimed that Definition 1 extends the semi-DP notion in Definition 2. However, this extension is not clearly stated in the current version. Specifically, I was confused about how to convert a special case of Definition 1 to Definition 2 (which part is public in Definition 1). Moreover, why is Definition 2 a stronger notion than Definition 1, as claimed by the authors?
>
>
> Thanks, we will add a detailed proof to how a semi-DP can be turned into a CCA mechansim in the final version.
> Roughly, every semi-DP mechanism $M$ with $k$ public datapoints defines a CCA mechanism $A$ as follows:
>
> For a dataset $S$, the CCA mechanism $A$ outputs $(h, R)$, where $R$ is the first $k$ datapoints in  $S$.
> The function $h = M(S_{\leq k}, S_{>k})$ is obtained by applying $M$ using the first $k$ examples in $S$ as public data and the remaining examples as private datapoints. Please let us know if further details are needed.
>
> ---
> We hope that our response addresses your concerns. Please let us know if we can answer any more questions!

---

### Official Review · Reviewer_jYet · 2024-07-13

**Soundness:** 3
**Presentation:** 3
**Contribution:** 3
**Rating:** 5
**Confidence:** 3

**Summary:**

This paper addresses the challenge of credit attribution within the context of machine learning algorithms. It proposes new definitions that relax the stability of a subset of data points. The framework extends established notions of stability, such as Differential Privacy, differentially private learning with public data, and stable sample compression, within the PAC learning framework. The authors provide a characterization of learnability for algorithms adhering to these stability principles and suggest future research directions.

**Strengths:**

1. The paper introduces novel definitions of stability that allow weaker stability of the designed subset of the data points.
2. This framework extends notions of stability, including Differential Privacy, differentially private learning with public data, and stable sample compression.

**Weaknesses:**

1. Examples that satisfy Definition 1 could be further discussed. Could the authors clarify examples of  (ε>0,δ)-counterfactual credit attributor (Definition 1) which are not (ε=0,δ=0)-CAA?
2. The conditional distribution in Definition 1 might make the definition challenging to verify.  It would enhance the paper if the authors could provide case studies of (ε, δ)-CAA applied to classical problems with varying parameters ε,δ controlling the privacy-utility tradeoff.

**Questions:**

1. See weaknesses
2. Could the authors explain the difference or connection between Definition 3 and adaptive composition + post-processing in DP?

---

> ### Author Rebuttal · Authors · 2024-08-07
>
> Thank you so much for reading our paper, and for your comments.
>
> With regards to addressing the weaknesses/questions:
>
> > Examples that satisfy Definition 1 could be further discussed. Could the authors clarify examples of $(\varepsilon>0,\delta)$-counterfactual credit attributor (Definition 1) which are not ($\varepsilon=0,\delta=0$)-CAA?
>
> Example 2.1 shows that any stable sample compression scheme satisfies Definition 1 with $\varepsilon=0, \delta=0$. In particular, Figure 1 shows that SVM satisfies $\varepsilon=0, \delta=0$. Furthermore, as we mention in the proof of Theorem 1, a slight variant of AdaBoost also satisfies this condition. This shows that even simple and well-known algorithms satisfy (a more restrictive version of) Definition 1.
>
> In terms of algorithms satisfying Definition 1 with $\varepsilon > 0$: any DP algorithm satisfies this condition, with $R=\emptyset$. Any semi-DP algorithm satisfies this condition, with $R \neq \emptyset$, but chosen non-adaptively (i.e., $R$ does not depend on the input). It would be interesting to construct examples of mechanisms that choose $R \neq \emptyset$ adaptively, and also satisfy Definition 1 with $\varepsilon > 0$. Note that we leave open (line 153) the question of constructing a PAC learner satisfying Definition 1 with $|R|=k=O(1)$, and conceivably, the first thing to try here might be to consider both $\varepsilon > 0$ and $R$ chosen adaptively.
>
> > The conditional distribution in Definition 1 might make the definition challenging to verify. It would enhance the paper if the authors could provide case studies of $(\varepsilon, \delta)$-CAA applied to classical problems with varying parameters $\varepsilon,\delta$ controlling the privacy-utility tradeoff.
>
> This is true, however this caveat is also true for DP and cryptographic security. It is not clear to us if there exists a verifiable definition in this context, but, like with DP, verifying Definition 1 will require an a priori proof by the algorithm designer.
>
> > Could the authors explain the difference or connection between Definition 3 and adaptive composition + post-processing in DP?
>
> Recall that the reconstruction function in Definition 3 is a semi-DP mechanism, and post-processing a semi-DP mechanism is also semi-DP. But if we compose a mechanism satisfying Definition 3, since the reconstruction function is a function of the $k$ compressed examples, privacy is foregone for these examples. For the points that do not get compressed, privacy decays as in advanced composition in DP.
>
> ---
> We hope our response addresses your concerns. Please let us know if we can answer any further questions!

---

> > ### Comment · Reviewer_jYet · 2024-08-14
> >
> > Thank you for your efforts in the rebuttal! I will maintain my original score.

---

### Official Review · Reviewer_JBKH · 2024-07-14

**Soundness:** 3
**Presentation:** 3
**Contribution:** 4
**Rating:** 7
**Confidence:** 3

**Summary:**

This paper studies the problem of credit attribution in machine learning tasks. Motivated by the moral and legal need to appropriately credit input data points when they significantly influence the output of a learning or generative model, the authors develop a characterization of reasonable credit attribution algorithms. Similar to how in differential privacy an individual is considered protected if keeping or omitting their data in the input data set leads to at most a limited $(\varepsilon,\delta)$ bounded shift in max-divergence of the output distribution, this work considers an individual to be safely not credited if omitting their data would have lead to a bounded shift in this sense.

The authors link the notion of generating a not-too-long list of samples to be credited (since a trivial solution is to credit everyone) to the notion of stable sample compression schemes of size $k$, which allow identification of a subsequence of size $k$ for every input data set such that restricting the input to any input subset containing that subsequence leads to the same output. This is stronger than the notion of credit attribution they are considering, so they consider a relaxation using semi-differentially private algorithms, which only need to have a bounded shift in the output distribution when a sample is added or dropped from a `private' subset of the input data. This relaxation is called a DP Sample Compression Scheme.

The authors give an example showing that their definition is not vacuous and go on to state three theorems:
1. PAC learnable classes with VC dimension $d$ admit solutions with valid credit attribution and only $O(d\log n)$ many samples.
2. If a concept class is not DP learnable (in which case it would have been automatically a valid credit attribution algorithm without having to credit any point), then it must credit at least $k=\Omega(1/\alpha)$ many points.
3. The additional number of points one is allowed to pick by relaxing from random subsampling of $k$ points (and crediting the whole sample) to a valid credit attribution algorithm allows one to pick at most an $\exp(\epsilon)$ factor many more samples.

**Strengths:**

1. This paper considers an important problem and comes up with a very reasonable definition to address it. Basing their notion of indistinguishability on that which is used in DP has found some mainstream and legal acceptance as well as far as I know, so in principle this might turn out to be quite viable.
2. The directions of inquiry (comparison with random subsampling, sample complexity, checking for potential vacuousity) are all good.

**Weaknesses:**

1. The notion of the privacy parameter $\varepsilon$ in standard DP has some interpretability in simple contexts like that of linear queries via lower bounds on the number of $\varepsilon$-private queries needed before reconstruction attacks become viable. The definition of credit attribution via DP necessitates some more thought into what a reasonable parameter choice looks like.
2. I feel strongly that the label 'DP Sample Compression Scheme' needs to be changed. To recall, in this paper's terms a DP Sample Compression Scheme is a mechanism $M: \mathcal{Z}^n \to \mathcal{C}$ such that $M(S) = \rho(S_{\kappa (S)},S_{\neg \kappa(S)})$, where $\kappa$ (the Compression function) is a $\varepsilon$-DP sampler from $S$, and $\rho$ (the Reconstruction function) is semi-DP (i.e. only DP with respect to its second component, the subset of points which were not sampled by $\kappa$). This is a good definition and makes sense, but I feel any label along the lines of DP <*> where <*> is a mechanism taking as input data sets suggests that it is in fact differentially private. This mechanism is clearly not DP (nor does it need to be for the purposes of this paper), because in particular any point sampled by $\kappa$ can be released in the clear (which is fine for the purposes of credit attribution). I understand that the subsampler is what is DP here, so maybe something along the lines of 'Sample Compression Scheme with DP compression or 'semi-DP sample compression scheme' might be a better descriptor, or whatever else the authors prefer.
One of the reasons I feel this is important is that we rely all the time on the post-processing properties of DP algorithms, and it is not unlikely that someone who did not read this definition properly mistakes this for a DP subroutine and uses it incorrectly as a DP blackbox in a larger DP algorithm. Any such instance would be that person's fault, but if we can reduce its likelihood/any sources of confusion that would be a good thing.

**Questions:**

If you can address the points I have raised in the Weaknesses section that would suffice for most of the questions I would like to have answered. In particular the second point is a bit concerning to me, although maybe I have misunderstood the definition. If my understanding is correct and if you feel strongly that the label/name ought to stay as it is and you can point to other important instances which violate the principle I described I would be happy to be corrected.

Additional questions/notes:
1. In the introduction you describe DP using swap DP but then later on use add-drop DP when defining credit attribution - perhaps you could just stick to add-drop DP throughout?
2. I think there might be an interesting link between user-level DP and appropriately attributing the works of an artist/data source, as opposed to any particular artwork/data point. I'm just curious if you have given this much thought, I might have missed it in the paper if so.

**Limitations:**

Yes, limitations have been adequately discussed.

---

> ### Author Rebuttal · Authors · 2024-08-07
>
> Thank you so much for reading our paper, and for your comments. We are glad you liked our paper!
>
> With regards to the points you bring up:
> > The notion of the privacy parameter $\varepsilon$ in standard DP has some interpretability in simple contexts like that of linear queries via lower bounds on the number of $\varepsilon$-private queries needed before reconstruction attacks become viable. The definition of credit attribution via DP necessitates some more thought into what a reasonable parameter choice looks like.
>
> This is a good point. We did not think much about the problem of credit attribution in terms of an indistinguishability test that the algorithm should pass under private queries; however, semantically mapping intuitions and interpretations from the now well-established theory on differential privacy to the setting of credit attribution seems like an excellent avenue for future research.
> > I feel strongly that the label 'DP Sample Compression Scheme' needs to be changed. To recall, in this paper's terms a DP Sample Compression Scheme is a mechanism $M:\mathcal{Z}^n \to \mathcal{C}$  such that $\mathcal{M}(S)=\rho(S_{\kappa(S)},S_{\neg \kappa(S)})$ where $\kappa$ (the Compression function) is a $\varepsilon$-DP sampler from $S$, and $\rho$ (the Reconstruction function) is semi-DP (i.e. only DP with respect to its second component, the subset of points which were not sampled by $\kappa$). This is a good definition and makes sense, but I feel any label along the lines of DP <*> where <*> is a mechanism taking as input data sets suggests that it is in fact differentially private. This mechanism is clearly not DP (nor does it need to be for the purposes of this paper), because in particular any point sampled by $\kappa$ can be released in the clear (which is fine for the purposes of credit attribution). I understand that the subsampler is what is DP here, so maybe something along the lines of 'Sample Compression Scheme with DP compression or 'semi-DP sample compression scheme' might be a better descriptor, or whatever else the authors prefer. One of the reasons I feel this is important is that we rely all the time on the post-processing properties of DP algorithms, and it is not unlikely that someone who did not read this definition properly mistakes this for a DP subroutine and uses it incorrectly as a DP blackbox in a larger DP algorithm. Any such instance would be that person's fault, but if we can reduce its likelihood/any sources of confusion that would be a good thing.
>
> This is a good point. We indeed mean that the compression function is DP but not further. We will consider the names Sample DP-Compression Scheme or Sample Compression Scheme with DP compression as suggested.
> > In the introduction you describe DP using swap DP but then later on use add-drop DP when defining credit attribution - perhaps you could just stick to add-drop DP throughout?
>
> Thanks, we will make this change.
> > I think there might be an interesting link between user-level DP and appropriately attributing the works of an artist/data source, as opposed to any particular artwork/data point. I'm just curious if you have given this much thought, I might have missed it in the paper if so.
>
> This is a great idea! If we understand your suggestion correctly, the principle of counterfactual attribution could also be stated at a per-user level, instead of a per-artwork level i.e., if in creating a work $W$, a mechanism does not cite/acknowledge any of the works $W_A$ by an author $A$, then the mechanism should be able to create $W$ as if it had not seen $W_A$ at all. This is a totally reasonable granularity to instantiate the definition, and constitutes an interesting formulation for future study.
>
> ---
> Please let us know if we can help answer any more questions!

---

> > ### Comment · Reviewer_JBKH · 2024-08-13
> > **Response to rebuttal**
> >
> > Thank you for the rebuttal, I think all of my questions and concerns have been addressed for now. My only main concern was about the term 'DP Sample Compression Scheme', and I'm sure that any term that you would like to use (not necessarily the ones I mentioned) should work well as long as it doesn't imply that the algorithm as a whole is DP.

---

### Author Rebuttal · Authors · 2024-08-07

We would like to thank the reviewers for taking the time to read the manuscript. We would like to reiterate that in this work, we present a first candidate notion of learning with credit attribution as well as provide a first characterization of PAC learnability under credit attribution. **This should be regarded as the scope of the current work.**

The reviewers have suggested several interesting and important open problems as well as future directions of research which we find all to be of great interest. We do want to emphasize, though, that while these suggestions highlight the potential of further study of credit attribution, our work is focused on presenting a model for credit attribution and initiates basic research of this model. We ask the reviewers to take this into account, and focus their assessment of our work as much as possible on the scope of this work and not on future study.

---

### Decision · Program_Chairs · 2024-09-25

**Decision:**

Accept (poster)

**Comment:**

The authors did a good job addressing the reviewers' questions and comments.  All reviewers are supportive. For this reason, I recommend accept.

Meanwhile, I do have some comments that the authors can consider.

First, on the technical level, the results appear to resemble the "semi-private learning" result very strongly. "Public data <=> data attribute credits to". Note that the most intriguing result on the dichotomy seems directly inherited from the ABM NeurIPS'19 paper. The discussion w.r.t. this line of work is somewhat ambiguous.

Secondly, it is philosophically and ethically a bit uneasy for me to accept that the protection of Definition 1 is sufficient. In the SVM example, sure, only the support vectors matter, but if you do not have the collection of all the other data points, how would you know that these data points that you are crediting are support vectors?

To drive the point home, let's say the company training this model is only providing monetary compensation to the support vectors, then seems very unfair to other data points that the algorithm actually used for training, and are very close to being the support vectors. I think the idea of conditioning on the output and finding a sparse number of data points that contribute to the output is a very broken paradigm for "credit attribution".